environmental engineering/materials science

graphene, nitrogen doping, desulfurization, density functional theory, hydration

**Authors for correspondence:**
Wei Chu
e-mail: chuwei1965@scu.edu.cn
Wanglai Cen
e-mail: cenwanglai@163.com

This article has been edited by the Royal Society of Chemistry, including the commissioning, peer review process and editorial aspects up to the point of acceptance.

# Enhanced heterogenous hydration of SO$_2$ through immobilization of pyridinic-N on carbon materials

Longhua Zou[1], Ping Yan[2], Peng Lu[4], Dongyao Chen[4], Wei Chu[1] and Wanglai Cen[1,3]

[1]Institute of New Energy and Low-carbon Technology, [2]College of Architecture and Environment, and [3]National Engineering Research Center for Flue Gas Desulfurization, Sichuan University, Chengdu 610065, People's Republic of China
[4]The Key Laboratory of Water and Air Pollution Control of Guangdong Province, South China Institute of Environmental Sciences, Ministry of Ecology and Environment of China, Guangzhou 510655, People's Republic of China

WChu, 0000-0002-7166-5443; WCen, 0000-0002-2854-964X

Carbon materials doped with nitrogen have long been used for SO$_2$ removal from flue gases for the benefits of the environment. The role of water is generally regarded as hydration of SO$_3$ which is formed through the oxidization of SO$_2$. However, the hydration of SO$_2$, especially on the surface of N-doped carbon materials, was almost ignored. In this study, the hydration of SO$_2$ was investigated in detail on the pyridinic nitrogen (PyN)-doped graphene (GP) surfaces. It is found that, compared with the homogeneous hydration of SO$_2$ assisted with NH$_3$ in gas phase, the heterogeneous hydration is much more thermodynamically and kinetically favourable. Specifically, when a single H$_2$O molecule is involved, the energy barrier for SO$_2$ hydration is as low as 0.15 eV, with 0.59 eV released, indicating the hydration of SO$_2$ can occur at rather low water concentration and temperature. Thermodynamic integration molecular dynamics results show the feasibility of the hydrogenated substrate recovery and the immobilized N acting as a catalytic site for SO$_2$ hydration. Our findings show that the heterogeneous hydration of SO$_2$ should be universal and potentially uncover the puzzling reaction mechanism for SO$_2$ catalytic oxidation at low temperature by N-doped carbon materials.

## 1. Introduction

The use of carbon-based materials for sulfur dioxide (SO$_2$) removal can be dated back to the 1980s [1–5]. It features the reaction of SO$_2$ in

the presence of $O_2$ and $H_2O$ which involves a series of reactions to produce sulfuric acid as the final product at relatively low temperature (20–150°C) conditions. It is conventionally recognized that $SO_2$ is oxidized by $O^*$ coming from the dissociation of adsorbed $O_2$ molecule to form $SO_3$ [6,7]. While the role of water is generally regarded as the hydration of $SO_3$ to form sulfuric acid and flush it out from the micropores of carbon materials [3,8]. The mechanism ignored the effects and participation of $H_2O$ in the early stages for $SO_2$ adsorption and its oxidation. However, since the concentration of water vapour in the flue gas is usually as high as 10 vol%, the hydration of adsorbed $SO_2$ on the carbon surface, especially N-doped carbon materials, is feasible. Furthermore, our previous published work [9] following the mechanism shows that an optimized energy barrier for $SO_2$ catalytic oxidation is in the range of 0.5–0.6 eV, which is still too high to explain the high performance of carbon materials for $SO_2$ removal at low temperature. There should be a new mechanism to address these issues, where the $H_2O$ effects should be taken into account in the early stages.

Substantial efforts have been dedicated to investigate the hydration of $SO_2$ in aqueous aerosols or gas phase both from experimental [10] and computational perspectives [11,12]. The adsorption and subsequent hydration process lead to the wet deposition of $SO_2$, which has a significant impact on the formation of sulfate aerosols [13]. It is noteworthy that 'sulfurous acid' ($H_2SO_3$) has never been characterized or isolated experimentally due to its short lifetime. It can be easily split into $SO_2$ and $H_2O$ [14]. Theoretical studies reveal that the gaseous hydration reaction of $SO_2$ without assistant is unfavourable. While the addition of water, $NH_3$ [15] or sulfuric acid [16] exhibits evident promotion effects by reducing the reaction barrier, due to a proton transmitter effect. Especially, the basic $NH_3$ molecule can promote the gaseous hydration of $SO_2$ both thermodynamically and kinetically, where $SO_2$ accepts $OH^-$ to form $HSO_3^-$ and $NH_3$ accepts $H^+$ to form $NH_4^+$. Recently, Lv *et al*. [17] reported that methylamine (MA) and dimethylamine (DMA) can also enhance $SO_2$ hydration with even lower energy barriers than ammonia. With only two $H_2O$ molecules involved, the DMA-assisted $SO_2$ hydration process is nearly barrierless. Inspired by the promotion effects of basic molecules for $SO_2$ hydration in gas phase or gas/water interface, we assume that the immobilization of basic pyridinic N groups on carbon materials can make the $SO_2$ hydration occur on the water–carbon interface in a heterogeneous way.

Actually, the incorporation of basic functional groups into carbon materials has been proved as an effective way to increase the adsorption and conversion of $SO_2$ [18,19]. Among them, the non-metallic nitrogen groups are most widely investigated for their high electron-donating capacity [20,21]. Therefore, the charge density and distribution of the original graphene (GP) lattice are modulated, inducing the formation of local activation regions [22,23] to enhance the interaction between carbon materials and polar molecules [24]. The nitrogen-containing groups mainly consist of pyridinic, pyrrolic and graphitic configurations. Among them, the pyridinic N (PyN), a N atom bonded with two C atoms and as a member of hexagon in GP lattice, has been experimentally confirmed the most basic one for introduction of strong electron donor states near the Fermi level [25,26]. And Raymundo-Piñero *et al*. [27] demonstrated that performance enhancement for $SO_2$ catalytic conversion is primarily related to the doping of PyN. Moreover, the doping of PyN is thermodynamically preferable [9], which usually occurs at the boundary of vacancy sites or at the edge of layers [28]. Through modulation of the C/N ratio of the precursors [25] and type of catalysts [29] during the preparation process, the pyridinic N is the predominant doping type to be obtained. Recently, a three PyN atoms-doped mono-vacancy structure has been synthesized and observed directly by scanning tunnelling microscopy (STM) [30], which makes it possible for precise tuning of PyN doping in the lattice of carbon-based catalysts.

Herein, we focused on the heterogeneous hydration of $SO_2$ on a range of PyN-doped GP substrates to uncover (i) the feasibility for the promotion of $SO_2$ hydration in thermodynamics and kinetics, (ii) the local structure of the active centre, and (iii) the hydration products and their implications for the total $SO_2$ catalytic oxidation. It is interesting to find that the hydration of $SO_2$ on PyN atoms-doped GP is much more preferable than the process in gaseous phase, indicating the important role of the water–carbon interface and a new reaction pathway for $SO_2$ catalytic oxidation.

## 2. Calculation details

A $3\sqrt{3} \times 6$ supercell of GP consisting of 72 C atoms was used as the basic model, with relaxed lattice parameters of $12.78 \times 14.76$ Å$^2$. A vacuum region of 15 Å was added perpendicular to the plane to avoid interaction between neighbouring layers. Mono- and di-vacancy were created in the plane of GP by removing one or two adjacent C atoms. Experimental [31] and our previous theoretical study [9]

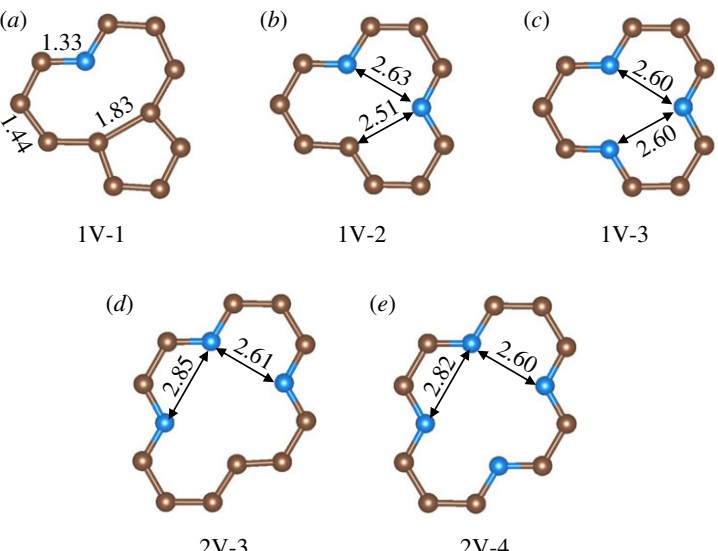

**Figure 1.** Local structures of relaxed pyridinic N-doped graphene models. Blue and brown spheres depict N and C atoms, respectively. Selected bond lengths are given in Å.

revealed that the unsaturated C atoms on the vacancy site of GP are inclined to be substituted by N atoms, namely substitution in the form of pyridinic N (PyN). When $x$ ($x = 1$–3) PyN atoms are introduced to the boundary of the mono-vacancy substrate, it is denoted as 1V-$x$. Following the same rule, the three and four PyN atoms doped di-vacancy substrate are denoted as 2V-3 and 2V-4, respectively (figure 1).

All the spin-polarized density functional theory (DFT) calculations were carried out using the Perdew–Burke–Ernzerhof (PBE) [32] functional with DFT-D3 correction [33] as implemented in the Vienna ab initio simulation package (VASP 5.4) [34,35]. A plane-wave basis set with energy cut-off of 400 eV was employed within the framework of the projector-augmented wave (PAW) method [36]. The Brillouin zone was sampled using a Monkhorst–Pack $3 \times 3 \times 1$ k-points mesh. Gaussian smearing with a smearing width of 0.2 eV was used. All the atoms were allowed to relax until the maximum Hellman–Feynman force on each atom was less than $0.02 \, \text{eV} \, \text{Å}^{-1}$, except the atoms on the boundary which were fixed in all directions. All the parameters used have been validated with reasonable accuracy in our previous works [6,37,38]. Climbing image nudged elastic band (CI-NEB) [39,40] method was employed to trace the minimum energy pathways (MEP) from an initial state (IS) to its final state (FS). The transition state (TS) was confirmed with a single imaginary frequency. The zero-point energy (ZPE) corrections were included for all energetic analysis. The adsorption energy $\Delta E_{\text{ads}}$ is defined as

$$\Delta E_{\text{ads}} = E_{\text{tot}} - (E_{\text{mol}} + E_{\text{sub}}),$$

where the $E_{\text{tot}}$, $E_{\text{mol}}$ and $E_{\text{sub}}$ are the total energies of the adsorption complex, isolated molecule and PyN-doped GP substrate, respectively.

For thermodynamic integration [41] calculations, the reaction trajectory was firstly explored by the metadynamic [42] method. Then, 10 ps constrained molecular dynamics simulation was carried out for each selected image. Canonical ensemble (with constant particles number N, volume V, and temperature T) at 300 K was used with timestep of 1 fs. The mass of H atom was set to 2 arb. units. Refer to electronic supplementary material for detailed parameters and description.

# 3. Results and discussion

## 3.1. Adsorption of SO₂ and H₂O

As shown in figure 2a, when a single PyN atom is introduced (1V-1), the SO₂ molecule is inclined to be parallel to the base plane, with the S atom atop of the PyN atom with a height of 3.13 Å. When a second PyN atom is added (1V-2), the adsorption configuration almost remains the same but with a shorter height of 2.81 Å. If more PyN atoms are introduced, as shown in 1V-3, 2V-3 and 2V-4, the adsorbed SO₂ molecule is drawn closer to the centre of the vacancy with the S atom slightly tilted down. The variation of the adsorption configurations and the monotonic increase in the adsorption energy

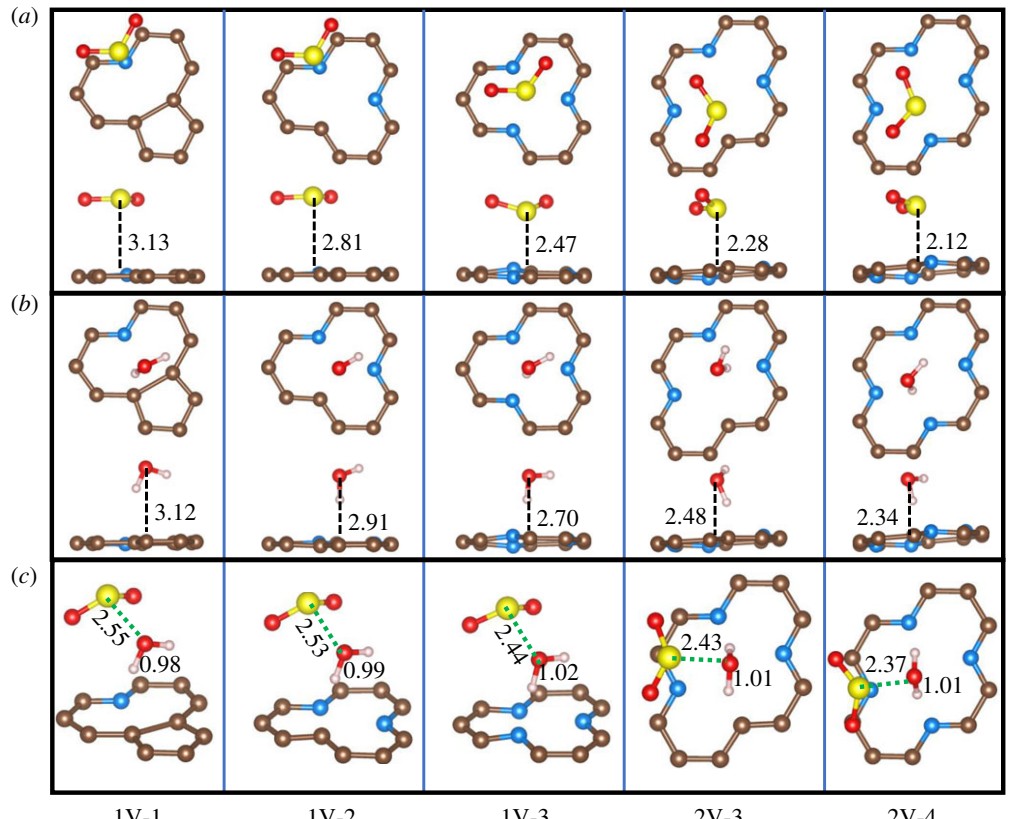

**Figure 2.** Adsorption configurations of $SO_2$ (*a*), $H_2O$ (*b*) and their co-adsorption (*c*) on different PyN-doped structures. The yellow, red and light pink balls depict S, O and H atoms, respectively. The others are depicted in the same way as figure 1. The distances are given in Å.

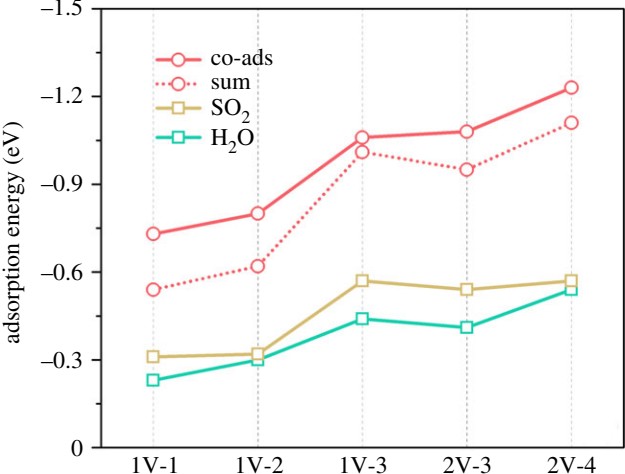

**Figure 3.** Adsorption energy of $H_2O$ and $SO_2$ on different PyN-doped structures. The 'sum' is simply the summation of the adsorption energy for separate $H_2O$ and $SO_2$ molecules.

(figure 3) with the increase of PyN atoms indicate that the adsorption of $SO_2$ can be enhanced by PyN doping. It should be due to the Lewis basicity of PyN sites and the more PyN sites, the stronger the basicity [43]. For the adsorption of $H_2O$ (figure 2*b*), in each of the configurations, the $H_2O$ molecule is adsorbed with one H atom pointing down to the vacancy. Following the same trend for $SO_2$ adsorption, the adsorption distance decreases according to the increase of PyN atoms number. It implies that the vacancy acts as a centre with negative charges, which is subject to adsorbing $H_2O$ and $SO_2$ through static Coulomb interaction. However, the adsorption energy for $H_2O$ is lower than that for $SO_2$. It might be due to more positive charge of S atom in $SO_2$ than H in $H_2O$.

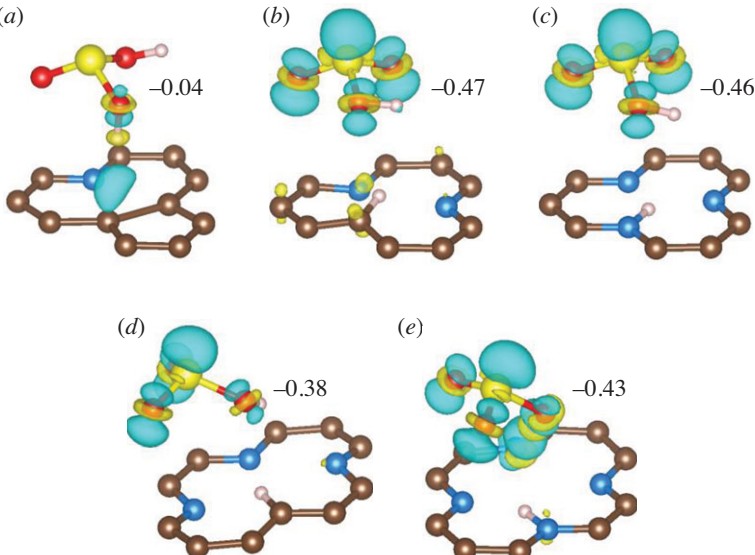

**Figure 4.** Charge difference and Bader population analysis of the products derived from $SO_2$ hydration on different PyN-doped structures. The isosurface is set to 0.1 eV Å$^{-3}$. The blue areas denote electron accumulation and yellow ones denote depletion.

For the co-adsorption of $H_2O$ and $SO_2$, considering the concentration of $H_2O$ (approx. 10 vol%) in real flue gases is higher than that of $SO_2$ (approx. 0.1 vol%) with an order of magnitude about 100, one $H_2O$ molecule was pre-absorbed before the introduction of $SO_2$. The adsorption configurations and energies are shown in figures 2c and 3, respectively. Geometrically, the adsorption distance from the S atom of $SO_2$ to the O atom of $H_2O$ is gradually shortened from 2.55 to 2.37 Å. Additionally, the bond length of the H–O pointing to the basal plane is elongated from 0.98 to 1.01–1.02 Å. They denote an increased interaction between $SO_2$ and $H_2O$ molecules, which may favour the hydration reaction of $SO_2$ by $H_2O$. The enhanced interaction can also be confirmed by the co-adsorption energy, which is higher than the sum of the separate adsorption energy of $SO_2$ and $H_2O$. This trend is consistent with experimental studies [44] that the nitrogen atoms in carbon materials behave as the main polar sites, which is beneficial for adsorption of polar molecules such as $H_2O$ and $SO_2$.

## 3.2. Products of $SO_2$ hydration

Figure 4 shows the structures and charge states of $SO_2$ hydration products based on several possible configurations tested (electronic supplementary material, figure S1). On 1V-1, the most possible product is $H_2SO_3$. For the other four situations, the thermodynamically feasible product is $HSO_3^{\delta-}$. Based on Bader population analysis, the value of $\delta$ is approximately 0.4–0.5. Considering the uncertainty of the Bader method and the geometry of $HSO_3^{\delta-}$, the product should be attributed to bisulfite. If more $H_2O$ molecules are included, which is practical in reaction conditions, they can help to stabilize $HSO_3^{\delta-}$ and the $\delta$ might be much closer to 1. Actually, when 7 and 24 $H_2O$ molecules are included, the calculated $\delta$ are 0.71 and 0.83, respectively (electronic supplementary material, figure S2 and table S1).

When $HSO_3^{\delta-}$ is formed, the H atom dissociated from $H_2O$ molecule can preferably be accepted by the unsaturated C atom (e.g. 1V-2 and 2V-3) of the PyN-doped substrates, then by one of the N atoms (e.g. 1V-3 and 2V-4). This kind of selectivity can be addressed that, for 1V-3 (figure 5c) and 2V-4 (figure 5e), the PyN atoms are the centres of negative charge and the highest occupied states of the substrate were localized on them, which is an object for H ion adsorption. For 1V-2 (figure 5b) and 2V-3 (figure 5d), the highest occupied states of the unsaturated C atom are on top of N atoms and are much closer to the Fermi level, resulting in the C site being more preferable than N for H acceptance.

## 3.3. Reaction pathway of $SO_2$ hydration

Figure 6 shows the energetics for the hydration reaction of $SO_2$ on different PyN-doped configurations corresponding to the products shown in figure 5. The hydration of $SO_2$ with single $H_2O$ molecule on 1V-1 is not preferable, with 0.31 eV cost. The corresponding barrier is as high as 1.10 eV, which is comparable to the value for $SO_2$ hydration in gas phase [15,45,46]. The kinetically unpreferable

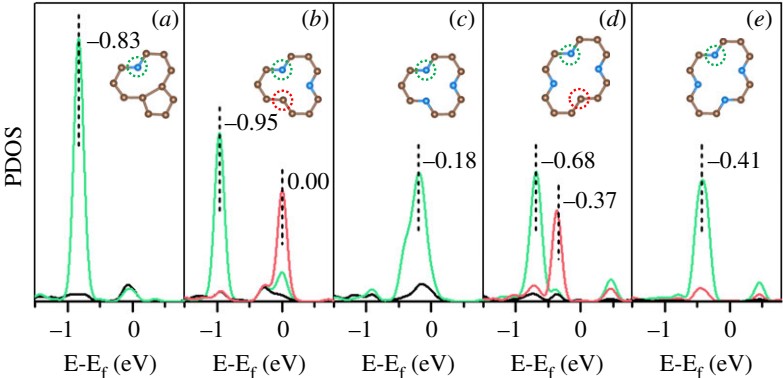

**Figure 5.** The highest occupied projected density of states (PDOS) for different PyN-doped substrates. (*a*) 1V-1, (*b*) 1V-2, (*c*) 1V-3, (*d*) 2V-3 and (*e*) 2V-4. The energy of Fermi level ($E_f$) was reset to 0 eV, and the value of $E-E_f$ refers to a relative energy to Fermi level. Corresponding N or C atoms are circled with dashed line in the same colour. States of an irrelated C atom far from the vacancy is also presented (black line) as reference.

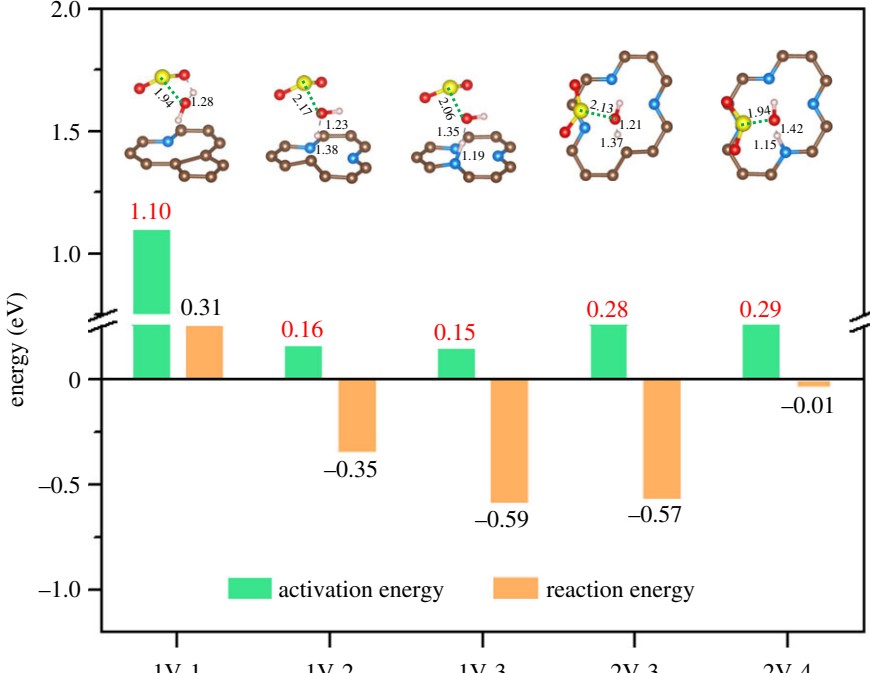

**Figure 6.** Reaction barriers (green bar) and energies (yellow bar) for $SO_2$ hydration on different PyN-doped substrates. Local structures of the transition states (TS) are included as insets.

hydration can be explained by the local structure of its transition states (TS) since a four-membered ring S–O–H–O with high inner tension was formed [15].

For the other four substrates, it is interesting to note that, all the hydration processes are thermodynamically preferable, with a certain energy released. Additionally, all the barriers for $SO_2$ hydration decrease to a value below 0.30 eV. Specifically, the energy barrier is 0.15 eV on 1V-3, which is lower than that assisted with ammonia [15] (0.54 eV, transferred from 12.53 kcal mol$^{-1}$) and MA (0.32 eV, transferred from 7.41 kcal mol$^{-1}$) and DMA [17] (0.21 eV, transferred from 4.78 kcal mol$^{-1}$) in the gas phase. The much lower barriers should be due to the hydration process following a different reaction pathway, where the $H_2O$ molecule is dissociating with one H atom approaching to the C or N atom of the substrate, meanwhile, the remaining OH reaching the S atom of $SO_2$ to form bisulfite as the product. In summary, when the number of doped PyN atoms is more than one, the reaction of hydration of $SO_2$ is feasible, both thermodynamically and kinetically.

Geometrically, the four feasible TS configurations can be categorized into two classes: the earlier TS class including 1V-2 and 2V-3, where the O–H bond is on the way to be broken and the H atom will be accepted by the unsaturated C atom, with H–C distance of 1.38 and 1.37 Å; and the latter TS class

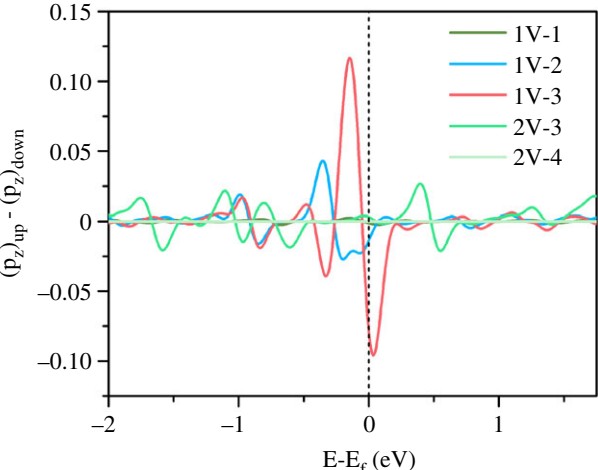

**Figure 7.** Spin-polarized $2p_z$ of the highest occupied states of different PyN-doped substrates: N for 1V-1, 1V-3, 2V-4 and C for 1V-2 and 2V-3. The Fermi level is set to 0 eV.

including 1V-3 and 2V-4, where the O–H bond has been completely broken and the H atom has been accepted by N atom, with a shorter H–N distance of 1.19 and 1.15 Å. The electronic local function (ELF, electronic supplementary material, figure S3) results denote stronger covalent interaction of H–N (0.93 for 2V-3 and 0.95 for 2V-4) than that of H–C (0.89 for 1V-2 and 0.90 for 2V-3).

However, the H atom acceptor C or N does not explain the inconsistent hydration barrier. For 2V-3 and 2V-4, they have different H acceptant, but share almost the same hydration barrier (0.28 and 0.29 eV). The same situation is found for 1V-2/3, but with much lower hydration barrier (0.16 and 0.15 eV). It should be partially addressed by the stronger co-adsorption of $SO_2$ and $H_2O$ on 2V-3/4 than on 1V-2/3 for the larger vacancy in size of the former, according to the adsorption energy results shown in figure 3.

Furthermore, it should be noted the barrier for $SO_2$ hydration increases in the order 1V-3 < 1V-2 < 2V-3 < 2V-4 < 1V-1. Comparing with the centre position of the highest occupied states in PDOSs 1V-2 > 1V-3 > 2V-3 > 2V-4 > 1V-1 (figure 5), the two series qualitatively match well except for 1V-2 and 1V-3. Correspondingly, figure 7 shows the spin-polarized $p_z$ states of all the highest occupied states. It was calculated directly by subtracting the spin-down states from the spin-up states. A higher spin-polarized $p_z$ state should be more feasible to accept H ion through covalent interaction, which can help to reduce the barrier of $SO_2$ hydration through $H_2O$ dissociation. There are three intensive signals just below the Fermi level in the order 1V-3 > 1V-2 > 2V-3, indicating the lowest $SO_2$ hydration barrier on 1V-3 and remedying the increase order 1V-3 < 1V-2.

Hence, we infer that the doping of negatively charged PyN atoms and the spin polarization states of C or N atoms cooperatively control the activity of PyN-doped substrates for $SO_2$ hydration. Compared with 2V-3/4, the electrons around the vacancy of 1V-2/3 are extruded and split, leading to much higher occupied C/N 2p states at the energy level and extra unpaired electrons of C $2p_z$ and N $2p_z$ perpendicular to the plane of the substrates. Therefore, the introduction of basic PyN groups can modulate the in-plane local electronic structure and enhances its electrostatic force to adsorb polar $H_2O$ and $SO_2$ molecules, while the asymmetric spin density out of plane dominates the chemical interaction and acts as selective sites for GP functionalization with proton coming from the dissociation of $H_2O$. The two factors result in the most active centre with three PyN atoms-doped mono-vacancy structure 1V-3 for $SO_2$ hydration.

## 3.4. Effects of extra water molecules

Since $H_2O$ is one of the majorities in fossil fuel burned flue gases and its presence is essential for $SO_2$ catalytic oxidation by nitrogen-doped carbon materials at low temperature, the role of $H_2O$ should not be limited to an absorbent of the product $SO_3$ as proposed conventionally. Here, we found the $SO_2$ can be hydrated feasibly to form bisulfite, which might be readily oxidized in further steps to form sulfuric acid. The oxidation pathway has been suggested recently based on experiment results on water microdroplets [47,48]. To generalize our findings, the effects of $H_2O$ for $SO_2$ hydration should be discussed in two points: (i) what are the effects if more $H_2O$ molecules present for $SO_2$ hydration on PyN-doped substrate, and (ii) can the hydrogenated substrates be dehydrogenated to recycle as a catalyst?

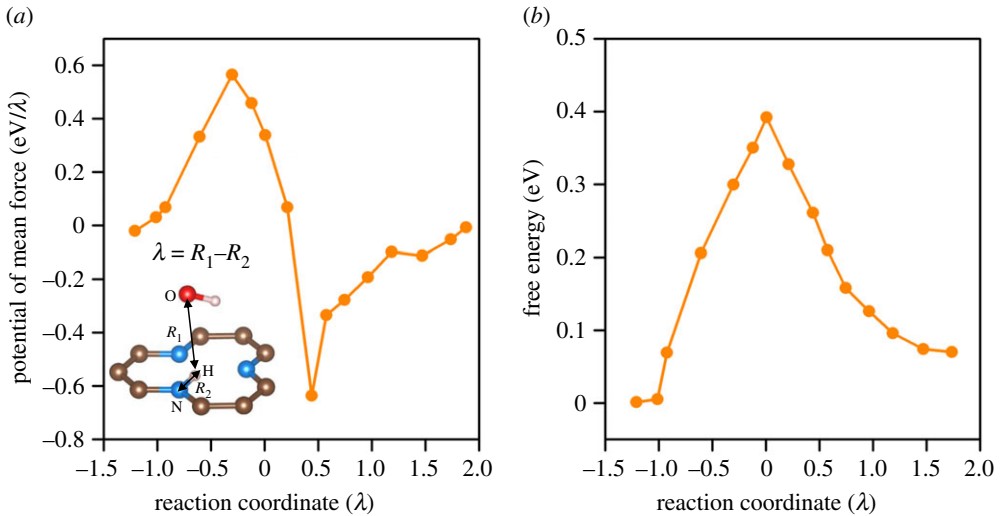

**Figure 8.** Potential of mean force (PMF) (*a*) and free energy profiles (*b*) along the reaction coordinate for dissociation of H ion from the 1V-3 substrate. The collective variable is chosen as $\lambda = R_1 - R_2$, where $R_1$ and $R_2$ is the distance of H–O and N–H as shown in the inset of (*a*).

Firstly, we propose that the hydration will be enhanced, specifically by reducing the barrier and increasing the total energy released to a limit. This trend has been reported for gaseous hydration reaction of $SO_2$ assisted by $NH_3$ and its derivatives of MA and DMA as mentioned before. For $NH_3$, the barrier for $n = 2$ is about 0.25 eV (transferred from 5.9 kcal mol$^{-1}$ in [13] and it is consistent with our calculated value *ca* 0.24 eV as shown in electronic supplementary material, figure S4) and it begins to release energy till $n = 3$. It is the same situation for amine, only when the number of water molecules is equal to or more than two, the hydration can be exothermic. For more $H_2O$ molecules involved in the reaction, the inner tension of the TS structure can be reduced to promote the hydration process. However, even when a single $H_2O$ molecule is included, the barrier is 0.15 eV on 1V-3, with 0.59 eV released. On 2V-4, although the reaction energy is irrelevant −0.01 eV, the value will increase to −0.06 and −0.31 eV when extra one and six $H_2O$ molecules are added, respectively (electronic supplementary material, figure S5). This is probably because the asymmetric spin density out of plane can significantly reduce the ring structure tension, thus the TS is more stable. On the other hand, this kind of enhancement effects both on barrier decrease and reaction energy increase can be attributed to the nature of the interface between PyN-doped carbon surface and the adsorbed $H_2O$ cluster. From the perspective of electronic structure, the introduction of pyridinic N atoms into the GP lattice changes the in-plane charge distribution around the dopant site. This leads to weak electrostatic forces, which promote the adsorption of polar molecules $H_2O$. A water cluster is inclined to be adsorbed on the PyN-doped site for its electronegativity and polarity through hydrogen bonding interaction, then facilitate the dissociation of $H_2O$ for $SO_2$ hydration as $NH_3$ and amine does in gaseous phase.

In order to investigate the dehydrogenation feasibility of the substrate after $SO_2$ hydration reaction, the free energy profile of the dehydrogenation of the hydrogenated pyridinic N was investigated with thermodynamic integration methods (refer to electronic supplementary material for details). Firstly, water dissociation was observed during the metadynamics simulation with Gaussian hills applied (electronic supplementary material, figure S6). Subsequently, thermodynamic integration calculations show that the $H_2O$ molecule dissociation process is endothermic (figure 8), indicating the reverse process for recovery of the hydrogenated substrate is thermodynamically preferable. This kind of dehydrogenation should be controlled by the influence of entropy [47], with a free energy decrease *ca* 0.1 eV and free energy barrier *ca* 0.3 eV, which are both kinetically and thermodynamically preferable. The simulation results indicate that the protonated substrate can be dehydrogenated and recovered for the next cycle of reaction.

## 4. Conclusion

Based on density functional theory calculations, it was found that $SO_2$ can be hydrated readily on the surface of pyridinic N (PyN)-doped carbon materials. Compared with the process in the gaseous phase, the gas–solid interface enhances the hydration of $SO_2$, both thermodynamically and kinetically.

The hydrated product can be attributed to bisulfite ($HSO_3^-$). Both the unsaturated C atoms or PyN atoms can act as active sites to accept H atoms to be hydrogenated, depending on their energy level of the highest occupied states and spin polarization. The most active sites should be the three PyN-doped mono-vacancy, with an energy barrier as low as 0.15 eV. When more $H_2O$ molecules are included, the heterogenous hydration of $SO_2$ can be further facilitated, and the hydrogenated sites can be recovered for the next hydration reaction. Our work shed insights on the important role of the water–carbon interface for $SO_2$ hydration and potentially opens a new reaction pathway for $SO_2$ catalytic oxidation.

Data accessibility. Our data are available from the Dryad Digital Repository: https://dx.doi.org/10.5061/dryad.6hdr7sqwr [49].

Authors' contributions. W.C. and W.C. designed the work and directed the experiments and supervised the project; L.Z. carried out the simulations and wrote the paper; P.Y. assisted the calculations; P.L. co-supervised the project; D.C. contributed to the manuscript revision; and all authors discussed the results and commented on the manuscript.

Competing interests. We declare we have no competing interests.

Funding. This work was funded by the Natural Science Foundation of China (grant no. 51878424) and the Key Laboratory of Water and Air Pollution Control of Guangdong Province, China (grant no. 2017A030314001).

Acknowledgement. The authors also acknowledge the Institute of New Energy and Low Carbon Technology of Sichuan University for computational support.

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
