## [Reviewer comments · Royal Society Open Science]

Review History

RSOS-192248.R0 (Original submission)

Review form: Reviewer 1

Is the manuscript scientifically sound in its present form?

No

Are the interpretations and conclusions justified by the results?

Yes

Is the language acceptable?

No

Do you have any ethical concerns with this paper?

No

Have you any concerns about statistical analyses in this paper?

No

Recommendation?

Reject

Comments to the Author(s)

Your manuscript entitled „Enhanced Heterogenous Hydration of SO₂ Through Immobilization of Pyridinic-N on Carbon Materials” presents the results of theoretical calculations on the titel reaction which can be considered as an addition to the calculations published by your group 4 years ago (Li, J., et al., Promotion mechanism of pyridine N-doped carbocatalyst for SO₂ oxidation. RSC Advances, 2016. 6(89): p. 86316-86323). That is why I have doubts whether the publication of this manuscript is now fully justified. If The Editor will decide to publish your manuscript I will accept his decision. In spite of the fact that I am not in a position to judge properly the language of the manuscript I am convinced that there is a room to improve its quality. For example, in my opinion the term „hydrolysis” is not appropriate to name the chemical interaction between SO₂ and H₂O. Therefore, you should use always the term „Hydration”. Moreover, the term „Especially, the alkaline NH₃ molecule” should be replaced by „basic NH₃.....”

Review form: Reviewer 2

Is the manuscript scientifically sound in its present form?

Yes

Are the interpretations and conclusions justified by the results?

Yes

Is the language acceptable?

Yes

Do you have any ethical concerns with this paper?

Yes

Have you any concerns about statistical analyses in this paper?

No

Recommendation?

Accept as is

Comments to the Author(s)

Overall, this work was well presented with good design of the experiments. I suggest to accept for publication as is.

Review form: Reviewer 3

Is the manuscript scientifically sound in its present form?

No

Are the interpretations and conclusions justified by the results?

No

Is the language acceptable?

Yes

Do you have any ethical concerns with this paper?

No

Have you any concerns about statistical analyses in this paper?

Yes

Recommendation?

Major revision is needed (please make suggestions in comments)

Comments to the Author(s)

The present manuscript titled: Enhanced Heterogeneous Hydration of SO₂ Through Immobilization of Pyridinic-N on Carbon Materials by Zou et al, shows the density functional theory calculations on the heterogeneous hydration of SO₂ on a range of PyN doped graphene substrates. They stated that their results shows the feasibility for the promotion of SO₂ hydration in thermodynamics and kinetics, the local structure of the active center and the hydration products and their implications for the total SO₂ catalytic oxidation. The work is interesting the manuscript suffers from a lot of contradictory statements. Some of the major comments are :

- The reason to choose pyridinic-N carbon materials is not clear. Authors should explain the superiority and the properties of this material with the help of recent references. What is the advantage of the present material compared to various materials reported in the study?
- The role of water-carbon interface should be explained in much more detail.
- The dehydrogenation feasibility of the substrate explanation should be rewritten in order to avoid the confusion to the readers.
- Active site comparison should also be done with the literature values
- They have to compare their results and work with the literature reports and have to show in which way this work is superior or better to the other research works. This work suffers with the novelty.

Decision letter (RSOS-192248.R0)

Dear Professor Cen:

Title: Enhanced Heterogeneous Hydration of SO₂ Through Immobilization of Pyridinic-N on Carbon Materials
Manuscript ID: RSOS-192248

The editor assigned to your manuscript has now received comments from reviewers. We would like you to revise your paper in accordance with the referee and Subject Editor suggestions which can be found below (not including confidential reports to the Editor). Please note this decision does not guarantee eventual acceptance.

I apologise that this has taken longer than usual; there was an issue with one of the reviewer reports which has now been resolved.

Please submit your revised paper before 05-Jun-2020. Please note that the revision deadline will expire at 00.00am on this date. If we do not hear from you within this time then it will be assumed that the paper has been withdrawn. In exceptional circumstances, extensions may be possible if agreed with the Editorial Office in advance. We do not allow multiple rounds of revision so we urge you to make every effort to fully address all of the comments at this stage. If deemed necessary by the Editors, your manuscript will be sent back to one or more of the original reviewers for assessment. If the original reviewers are not available we may invite new reviewers.

RSC Associate Editor:
Comments to the Author:
(There are no comments.)

RSC Subject Editor:
Comments to the Author:
(There are no comments.)

Reviewers' Comments to Author:
Reviewer: 1

Comments to the Author(s)
Your manuscript entitled „Enhanced Heterogenous Hydration of SO₂ Through Immobilization of Pyridinic-N on Carbon Materials” presents the results of theoretical calculations on the titel reaction which can be considered as an addition to the calculations published by your group 4

years ago (Li, J., et al., Promotion mechanism of pyridine N-doped carbocatalyst for SO₂ oxidation. RSC Advances, 2016. 6(89): p. 86316-86323). That is why I have doubts whether the publication of this manuscript is now fully justified. If The Editor will decide to publish your manuscript I will accept his decision. In spite of the fact that I am not in a position to judge properly the language of the manuscript I am convinced that there is a room to improve its quality. For example, in my opinion the term „hydrolysis“ is not appropriate to name the chemical interaction between SO₂ and H₂O. Therefore, you should use always the term „Hydration“. Moreover, the term „Especially, the alkaline NH₃ molecule“ should be replaced by „basic NH₃.....“

Reviewer: 2

Comments to the Author(s)

Overall, this work was well presented with good design of the experiments. I suggest to accept for publication as is.

Reviewer: 3

Comments to the Author(s)

The present manuscript titled: Enhanced Heterogeneous Hydration of SO₂ Through Immobilization of Pyridinic-N on Carbon Materials by Zou et al, shows the density functional theory calculations on the heterogeneous hydration of SO₂ on a range of PyN doped graphene substrates. They stated that their results shows the feasibility for the promotion of SO₂ hydration in thermodynamics and kinetics, the local structure of the active center and the hydration products and their implications for the total SO₂ catalytic oxidation. The work is interesting the manuscript suffers from a lot of contradictory statements. Some of the major comments are :

- The reason to choose pyridinic-N carbon materials is not clear. Authors should explain the superiority and the properties of this material with the help of recent references. What is the advantage of the present material compared to various materials reported in the study?
- The role of water-carbon interface should be explained in much more detail.
- The dehydrogenation feasibility of the substrate explanation should be rewritten in order to avoid the confusion to the readers.
- Active site comparison should also be done with the literature values
- They have to compare their results and work with the literature reports and have to show in which way this work is superior or better to the other research works. This work suffers with the novelty.

Author's Response to Decision Letter for (RSOS-192248.R0)

See Appendix A.

RSOS-192248.R1 (Revision)

Review form: Reviewer 1

Is the manuscript scientifically sound in its present form?

Yes

Are the interpretations and conclusions justified by the results?

Yes

Is the language acceptable?

Yes

Do you have any ethical concerns with this paper?

No

Have you any concerns about statistical analyses in this paper?

No

Recommendation?

Accept as is

Comments to the Author(s)

After revision, your manuscript can be published

Decision letter (RSOS-192248.R1)

Dear Professor Cen:

Title: Enhanced Heterogenous Hydration of SO₂ Through Immobilization of Pyridinic-N on Carbon Materials

Manuscript ID: RSOS-192248.R1

It is a pleasure to accept your manuscript in its current form for publication in Royal Society Open Science. The chemistry content of Royal Society Open Science is published in collaboration with the Royal Society of Chemistry.

RSC Associate Editor:
Comments to the Author:
(There are no comments.)

RSC Subject Editor:
Comments to the Author:
(There are no comments.)

Reviewer(s)' Comments to Author:
Reviewer: 1

Comments to the Author(s)
After revision, your manuscript can be published

Appendix A

Response to the editor and reviewers

Dear Dr Laura Smith and reviewers,

Thank you for your letter and the reviewers' comments concerning our manuscript entitled "Enhanced Heterogenous Hydration of SO₂ Through Immobilization of Pyridinic-N on Carbon Materials", Manuscript ID: RSOS-192248. We have carefully studied and followed these constructive suggestions, and made it revised wherever it is available. The details of our responses to the suggestions are listed point by point as below. All changes in manuscript have been highlight in red font.

We sincerely hope that the revised manuscript and our accompanying responses will be sufficient to make our manuscript suitable for publication in Royal Society Open Science. Thank you again for your assistance with our manuscript!

Best regards,

Dr. Wanglai Cen

Institute of New Energy and Low Carbon Technology

Sichuan University, China.

Reviewer #1:

Your manuscript entitled “Enhanced Heterogenous Hydration of SO₂ Through Immobilization of Pyridinic-N on Carbon Materials” presents the results of theoretical calculations on the title reaction which can be considered as an addition to the calculations published by your group 4 years ago (Li, J., et al., Promotion mechanism of pyridine N-doped carbocatalyst for SO₂ oxidation. RSC Advances, 2016. 6(89): p. 86316-86323). That is why I have doubts whether the publication of this manuscript is now fully justified. If the Editor will decide to publish your manuscript, I will accept his decision. In spite of the fact that I am not in a position to judge properly the language of the manuscript I am convinced that there is a room to improve its quality. For example, in my opinion the term “hydrolysis” is not appropriate to name the chemical interaction between SO₂ and H₂O. Therefore, you should use always the term “Hydration”. Moreover, the term “Especially, the alkaline NH₃ molecule” should be replaced by “basic NH₃.....”

Response: Thanks for the comment and suggestions. Firstly, the inappropriate words “hydrolysis” and “alkaline” have been corrected accordingly. We have gone through the manuscript and several other words or expressions have been revised as well.

Secondly, there could be some misunderstanding regarding the present work due to our improper words or expressions. We think that it is a new work and quite different from the published one as the reviewer mentioned (Li, J., et al., Promotion mechanism of pyridine N-doped carbocatalyst for SO₂ oxidation. RSC Advances, 2016. 6(89): p. 86316-86323). The mechanism for SO₂ removal in the presence of H₂O and O₂ catalyzed by carbon materials has generally been regarded consisting three steps[1, 2]: (1) O₂ molecule adsorption and dissociation to form surface active O*, (2) SO₂ oxidation by O* to form adsorbed SO₃, and (3) SO₃ hydration by H₂O to form sulfuric acid. The mechanism which was used in the published work ignored the effects and participation of H₂O in the early stages for SO₂ adsorption and its oxidation. However, since the fraction of H₂O as a component of SO₂ contained tail gas can be up to 10

vol %, the hydration of adsorbed SO₂ on the carbon surface, especially N doped carbon materials, is feasible. Furthermore, Li's published work shows that an optimized energy barrier for SO₂ catalytic oxidation is in the range of 0.5~0.6 eV, which is still too high to explain the high performance of carbon materials for SO₂ removal at low temperature. There should be a new mechanism to address these issues, where the H₂O effects should be taken into account in the early stages. This is the motivation of the present manuscript.

As mentioned in the manuscript, the promotion of basic ammonia to SO₂ hydration in gaseous phase inspires us to rethink the effects of H₂O in the earlier stages of SO₂ catalytic oxidation on carbon materials, specifically the heterogeneous hydration of SO₂. Through DFT calculations conducted in the present manuscript, we have found that the heterogeneous SO₂ hydration on pyridinic N doped carbon materials is much more preferable than that in gaseous phase. Therefore, it may be the hydronated SO₂ of (HSO₃)⁻ rather than SO₂ molecule to be oxidized. The follow-up study for oxidation of (HSO₃)⁻, the SO₂ hydrated product is in progress, and we find that the energy barrier of the reaction of (HSO₃)⁻ + O₂ → (HSO₅)⁻ is about 0.07 eV. Subsequently, we believe the present work is new, and quite different from any of our published work.

The principal additions and changes in the manuscript are listed as below:

Line 37-38 in Page 2: "It is conventionally recognized that SO₂ is oxidized by O* coming from the dissociation of adsorbed O₂ molecule to form SO₃."

Line 43-47 in Page 3: "Furthermore, our previous published work [9] following the mechanism show that an optimized energy barrier for SO₂ catalytic oxidation is in the range of 0.5~0.6 eV, which is still too high to explain the high performance of carbon materials for SO₂ removal at low temperature. There should be a new mechanism to address these issues, where the H₂O effects should be taken into account in the early stages."

Line 57-61 in Page 4: "Recently, Lv [17] etc. reported that the methylamine (MA) and dimethylamine (DMA) can also enhance SO₂ hydration with even lower energy barriers than ammonia. With only two H₂O molecules involved, the DMA assisted SO₂ hydration process is nearly barrierless. Inspired by the promotion effects of basic

molecules for SO₂ hydration in gas phase or gas/water interface”

Reviewer #2:

Overall, this work was well presented with good design of the experiments. I suggest to accept for publication as is.

Response: Thanks a lot for the reviewer’s positive comment on our work.

Reviewer #3:

The present manuscript titled: Enhanced Heterogenous Hydration of SO₂ Through Immobilization of Pyridinic-N on Carbon Materials by Zou et al, shows the density functional theory calculations on the heterogeneous hydration of SO₂ on a range of PyN doped graphene substrates. They stated that their results show the feasibility for the promotion of SO₂ hydration in thermodynamics and kinetics, the local structure of the active center and the hydration products and their implications for the total SO₂ catalytic oxidation. The work is interesting the manuscript suffers from a lot of contradictory statements. Some of the major comments are:

Comment 1: The reason to choose pyridinic-N carbon materials is not clear. Authors should explain the superiority and the properties of this material with the help of recent references. What is the advantage of the present material compared to various materials reported in the study?

Response: Thanks for the reviewer’s suggestion. In the case of SO₂ removal from the flue gasses, nitrogen containing porous carbon materials have shown excellent performance during SO₂ adsorption processes[1, 3, 4], owing to its basic properties. The PyN is the most basic type, compared with graphitic and pyrrolic ones. Theoretical studies have shown that the formation energy of PyN is much lower than graphitic N[5], implying the energetically feasibility of PyN doped carbon materials preparation in experiment. We have further clarified the reason for choosing pyridinic-N carbon

materials in the introduction section of the manuscript. References [25], [26], [27] and [29] have been added to the manuscript. The principal additions and changes in the manuscript are listed as below:

Line 70-74 in Page 4: “Among them, the pyridinic N (PyN), a N atom bonded with two C atoms and as a member of hexagon in graphene lattice, has been experimentally confirmed the most basic one for introduction of strong electron donor states near the Fermi level [25, 26]. And E. Raymundo-Pinero [27] etc. demonstrated that performance enhancement for SO₂ catalytic conversion is primarily related to the doping of PyN. Moreover, the doping of PyN”

Line 76-78 in Page 4: “Through modulation of the C/N ratio of the precursors[25] and type of catalysts[29] during the preparation process, the pyridinic N is the predominant doping type to be obtained.”

Comment 2: The role of water-carbon interface should be explained in much more detail.

Response: Thanks for the comment and suggestions. Firstly, the immobilization of basic pyridinic N groups on carbon materials can make the SO₂ hydration occurs on the water-carbon interface through a heterogeneous way. Moreover, the SO₂ hydration on 1V-3 is both kinetically and thermodynamically preferable than ammonia, methylamine and dimethylamine in the gas phase with only one H₂O molecule involved, which means that SO₂ hydration is readily to happen on the surface of PyN containing carbon materials under low temperature.

From the perspective of electronic structure, the introduction of pyridinic N atoms into the graphene lattice changes the in-plane charge distribution around the dopant site. This leads to weak electrostatic forces, which promotes the adsorption of polar molecules H₂O [6]. Through hydrogen bonding, water clusters can be formed around the N doping site. More water molecules are beneficial for SO₂ hydration and the formation of water clusters around the doping site can further promote SO₂ hydration

just as ammonia and amine does in the gas phase.

The principal additions and changes in the **manuscript** are listed as below:

Line 61-63 in Page 4: “the immobilization of basic pyridinic N groups on carbon materials can make the SO₂ hydration occurs on the water-carbon interface through a heterogeneous way.”

Line 262-265 in Page 16: “From the perspective of electronic structure, the introduction of pyridinic N atoms into the graphene lattice changes the in-plane charge distribution around the dopant site. This leads to weak electrostatic forces, which promotes the adsorption of polar molecules H₂O.”

Comment 3: The dehydrogenation feasibility of the substrate explanation should be rewritten in order to avoid the confusion to the readers.

Response: Thanks for the critical and nice suggestion. We have redrawn Fig. 8 and revised context accordingly in the manuscript as below.

Line 269-270 in Page 16: “the free energy profile of the dehydrogenation of the hydrogenated pyridinic-N was investigated with thermodynamic integration methods (refer supporting information for details).”

Line 276-278 in Page 17: “which are both kinetically and thermodynamically preferable. The simulation results indicate that the protonated substrate can be dehydrogenated and recovered for next cycle of reaction.”

Figure R1. Potential of mean force (PMF) (a) and free energy profiles (b) along the reaction coordinate for dissociation of H ion from the 1V-3 substrate. The collective variable is chosen as $\lambda = R_1 - R_2$, where R_1 and R_2 is the distance of H-O and N-H as shown in the inset of (a).

Comment 4: Active site comparison should also be done with the literature values.

Response: Thanks for the reviewer’s critical suggestion. The introduction of basic PyN groups creates active sites on the surface of carbon materials through modulation of the in-plane local electronic structure which leads to weak electrostatic forces, thus the adsorption of H_2O and SO_2 molecules are promoted and activated. While the asymmetric spin density out of plane dominates the chemical interaction and act as selective sites for graphene functionalization with the proton coming from the dissociation of H_2O . The active site is further illustrated in the manuscript as:

Line 232-236 in Page 14: “Therefore, the introduction of basic PyN groups can module the in-plane local electronic structure and enhances its electrostatic force to adsorb polar H_2O and SO_2 molecules. While the asymmetric spin density out of plane dominates the chemical interaction and act as selective sites for graphene functionalization with proton coming from the dissociation of H_2O .”

The promotion effect of immobilized basic PyN on SO₂ hydration was compared with that of basic ammonia and amines from perspectives of hydration energy barriers with only one H₂O molecule involved, as listed in the manuscript of:

Line 195-198 in Page 11: “Specifically, the energy barrier is 0.15 eV on 1V-3, which is lower than that assisted with ammonia [15] (0.54 eV, transferred from 12.53 kcal·mol⁻¹) and MA (0.32 eV, transferred from 7.41 kcal·mol⁻¹) and DMA [17](0.21 eV, transferred from 4.78 kcal·mol⁻¹) in the gas phase.”

Comment 5: They have to compare their results and work with the literature reports and have to show in which way this work is superior or better to the other research works. This work suffers with the novelty.

Response: Thanks for the reviewer’s suggestions. This work is an exploration of the SO₂ catalytic oxidization mechanism in the presence of H₂O on carbon materials. SO₂ is conventionally thought to be oxidized to form SO₃ first, then SO₃ is hydrated to form sulfuric acid. However, the role of H₂O in the early stages during SO₂ adsorption and oxidation process has been neglected, in spite of considerable amount of water vapor in the flue gas. Stimulated by the promotion of SO₂ hydration by basic ammonia and amine molecules in the gas phase, we propose the feasibility of SO₂ hydration on the PyN containing carbon surfaces. The DFT calculation results have confirmed the hypothesis, SO₂ hydration is both kinetically and thermodynamically preferable on PyN doped carbon materials. Taking into account the participation of H₂O in the early stage of SO₂ catalytic oxidization is the main purpose and novelty of the present manuscript.

Liu[7] and Lv[8] etc. have demonstrated that ammonia and amine can be actively involved in the hydrogen atom transfer reactions, thus the transition state structure can be stabilized through reducing the tension of the ring structure formed by SO₂ and H₂O. In spite of this, only when the number of water molecules is equal or more than two, the hydration can be exothermic. Compared with ammonia and amine in the gas phase, PyN doped carbon materials can preferably promote SO₂ hydration with lower energy

barrier and more reactive energy released with only one H₂O molecule involved. This is probably due to the asymmetric spin density out of plane can significantly reduce the inner tension of the ring structure, thus the transition state is more stable. The principal additions and changes in the manuscript are listed as below:

Line 253-255 in Page 15: “It is the same situation of amine, only when the number of water molecules is equal or more than two, the hydration can be exothermic. For more H₂O molecules involved in the reaction, the inner tension of the transition state structure can be reduced to promote the hydration process.”

Line 258-260 in Page 16: “This is probably due to the asymmetric spin density out of plane can significantly reduce the ring structure tension, thus the transition state is more stable.”

References

1. Qu, Z., et al., *The effect of nitrogen-containing functional groups on SO₂ adsorption on carbon surface: Enhanced physical adsorption interactions*. Surface Science, 2018. **677**: p. 78-82.
2. Lizzio, A.A. and J.A. DeBarr, *Mechanism of SO₂ Removal by Carbon*. Energy & Fuels, 1997. **11**(2): p. 284-291.
3. Abdurashheed, A.A., et al., *Surface modification of activated carbon for adsorption of SO₂ and NO_x: A review of existing and emerging technologies*. Renewable and Sustainable Energy Reviews, 2018. **94**: p. 1067-1085.
4. Zhang, H., et al., *Adsorption and oxidation of SO₂ by graphene oxides: A van der Waals density functional theory study*. Applied Surface Science, 2015. **324**: p. 61-67.
5. Li, J., et al., *Tailoring Active Sites via Synergy between Graphitic and Pyridinic N for Enhanced Catalytic Efficiency of a Carbocatalyst*. ACS Appl Mater Interfaces, 2017. **9**(23).
6. Mallada, B., et al., *Atomic-Scale Charge Distribution Mapping of Single Substitutional p- and n-Type Dopants in Graphene*. ACS Sustainable Chemistry & Engineering, 2020. **8**(8): p. 3437-3444.
7. Liu, J., et al., *Mechanism of the Gaseous Hydrolysis Reaction of SO₂: Effects of NH₃ versus H₂O*. The Journal of Physical Chemistry A, 2015. **119**(1): p. 102-111.
8. Lv, G., et al., *Towards understanding the role of amines in the SO₂ hydration and the contribution of the hydrated product to new particle formation in the Earth's atmosphere*. Chemosphere, 2018. **205**: p. 275-285.